# Development and application of LAMP assays for the detection of enteric adenoviruses in feces

Anna K. Shuryaeva,[a] Tatyana V. Malova,[a] Anna A. Tolokonceva,[a] Sofia A. Karceka,[a] Maria A. Gordukova,[b] Ekaterina E. Davydova,[a] German A. Shipulin[a]

aFederal State Budgetary Institution Centre for Strategic Planning and Management of Biomedical Health Risks of the Federal Medical Biological Agency, Moscow, Russian Federation
bG.N. Speranskiy Children Hospital No. 9, Moscow, Russian Federation

**ABSTRACT** Loop-mediated isothermal amplification (LAMP) is an alternative to PCR that is faster and requires fewer resources. Here, we describe two LAMP assays for the detection of human adenoviruses in the feces of children with acute intestinal infections. We designed colorimetric LAMP (c-LAMP) and real-time LAMP (f-LAMP) with fluorescent probes to detect the DNA of the adenovirus F human adenovirus 40/41 (hAdV40/41) hexon gene. The detection limit of both developed methods was $10^3$ copies/mL, which is comparable to the sensitivity of PCR. The specificities of both c-LAMP and f-LAMP were high, with no false-positive results for clinical samples that do not contain adenovirus F, when testing other viruses and microorganisms. Comparative tests of PCR and LAMP on clinical samples from patients with acute gastroenteritis were carried out. For all samples with a PCR threshold cycle ($C_T$) of up to 36, the PCR and LAMP results completely coincided; however, at low viral loads, the diagnostic sensitivity of LAMP, especially c-LAMP with colorimetric detection, was inferior to that of PCR. The combination of LAMP with modern methods of nucleic acid extraction, both in manual and automatic modes, can reduce the time for a complete study, including extraction of nucleic acid material and amplification, to 60 min.

**IMPORTANCE** In April 2022, several cases of acute hepatitis of unknown origin were reported in children from 12 countries. In many cases, enteric adenovirus or SARS-CoV-2 and adenovirus coinfection were detected. It is known that human adenoviruses can cause different infections of varying severity, from asymptomatic to severe cases with lethal outcomes. There is a need to increase the diagnostic capabilities of clinical laboratories to identify such an underestimated pathogen as adenovirus. Although PCR remains the gold standard for pathogen detection, this method requires specialized equipment and has a long turnaround time to process samples. Previously, LAMP assays for the detection of human adenovirus have been based on measuring the turbidity, the fluorescence of intercalated dyes, or electrophoretic separation. Herein, we present LAMP-based assays with colorimetric or fluorescent detection and perform a detailed assessment of their sensitivity, specificity, and diagnostic performance.

**KEYWORDS** loop-mediated isothermal amplification, LAMP, DNA extraction, molecular diagnostics, human adenoviruses 40 and 41, acute gastroenteritis

Address correspondence to Ekaterina E. Davydova, edavydova@cspmz.ru.

The authors declare no conflict of interest.

Acute gastroenteritis is one of the most common infections among children. The most important etiologic causes responsible for approximately 70% of the episodes of acute gastroenteritis in children are rotavirus (4 to 47%), norovirus (3 to 29%), adenovirus F (1 to 31%), and astrovirus (1.5 to 16%) (1–5).

Adenoviruses F human adenovirus 40 (hAdV40) and hAdV41 are enteric pathogens that are recognized as a leading cause of gastroenteritis and diarrhea-associated mortality in young children just behind rotaviruses (6–8).

Adenovirus infections have a long incubation period (typically 8 to 10 days), and the illness can be prolonged for as long as 2 weeks. The most common symptoms of adenovirus infections are vomiting, fever, and diarrhea (9).

The great burden of viral gastroenteritis on health care due to related illness and hospitalization highlights the need for fast, sensitive, and reliable diagnostic assays to guide infection control measures. The loop-mediated isothermal amplification (LAMP) assay has been successfully implemented for the detection of a wide range of pathogens of infectious diseases, including SARS-CoV-2 (10), Zika virus (11, 12), enteroviruses (13), Dengue virus (14), tuberculosis (15, 16), Ebola virus (17), influenza virus (18), *Helicobacter pylori* (19), *Plasmodium falciparum* (20), *Leishmania* (20), *Trypanosoma* (20), *Treponema pallidum* (21), and *Haemophilus ducreyi* (21). LAMP is a highly specific and sensitive reaction of DNA amplification, as LAMP significantly reduces the amplification time from 2 to 3 h to 15 to 30 min compared to that of PCR.

An important advantage of the LAMP assay is its simplicity. It does not require expensive thermocyclers since the reaction is performed at a constant temperature between 60°C and 65°C, and the detection of the results is possible in a simple visual way due to the enormous amount of white precipitate of magnesium pyrophosphate being produced as a by-product of the amplification or a change in the color of pH-sensitive dyes.

Well-equipped laboratories can use fluorescently labeled sequence-specific probes for LAMP product detection, and using different fluorescent dyes makes it possible to multiplex LAMP reactions by analogy with multiplex PCR.

LAMP assays were described for the amplification of enteric adenovirus F hAdV40 and hAdV41 hexon genes (22), with the detection of products by measuring the turbidity, the fluorescence change with intercalated dyes, the color change with pH-dependent dyes, or electrophoretic separation. However, the proposed methods are not specific and sensitive enough, and in some cases, electrophoretic detection is time-consuming.

In this work, colorimetric LAMP (c-LAMP) and real-time LAMP (f-LAMP) assays with fluorescently labeled specific probes were developed to detect the DNA of adenoviruses of group F hAdV40/41. The diagnostic characteristics of these methods were compared with each other and with the PCR assay used in clinical diagnostics practice.

## RESULTS

**Colorimetric LAMP and real-time LAMP with assimilating FRET primer probes.** LAMP primers were designed at the 5′ region of the hexon gene; this region is conserved throughout the human adenovirus F taxon and differs from all other adenoviruses, including human adenoviruses A, B, E, and D. Six LAMP primers targeting 8 regions for hAdV40/41 DNA amplification and assimilating fluorescence resonance energy transfer (FRET) primer probes for f-LAMP were designed.

The maximum number of database sequences of the human adenovirus F hAdV40/41 groups was BLAST screened to make sure that the sequences of primers have high identity to the target region of hAdV40/41 genomes.

Some single polymorphisms at the 3′ ends of the primers were accounted for by the degeneration of the primer sequence.

The primer specificity was first evaluated via sequence alignment with the hAdV1, hAdV2, hAdV3, hAdV4, hAdV5, hAdV6, hAdV7, hAdV8, and hAdV21 mastadenoviruses and the performance of a homology search using BLAST screening outside the hAdV40/41 groups.

To measure the analytical sensitivity of c-LAMP and f-LAMP, serial dilutions of pAdvF plasmid DNA containing a fragment of the *hexon* gene with concentrations ranging from $10^2$ copies/mL to $10^4$ copies/mL were used (Fig. 1 and 2).

In both assays, c-LAMP and f-LAMP with assimilating FRET primer probes allowed the detection of adenovirus DNA at a concentration of $10^3$ copies/mL or 10 copies per reaction, showing positive amplification for this concentration in all repeats. Thus, c-LAMP and f-LAMP showed high analytical sensitivity, comparable to PCR methods (Fig. 1 and 2).

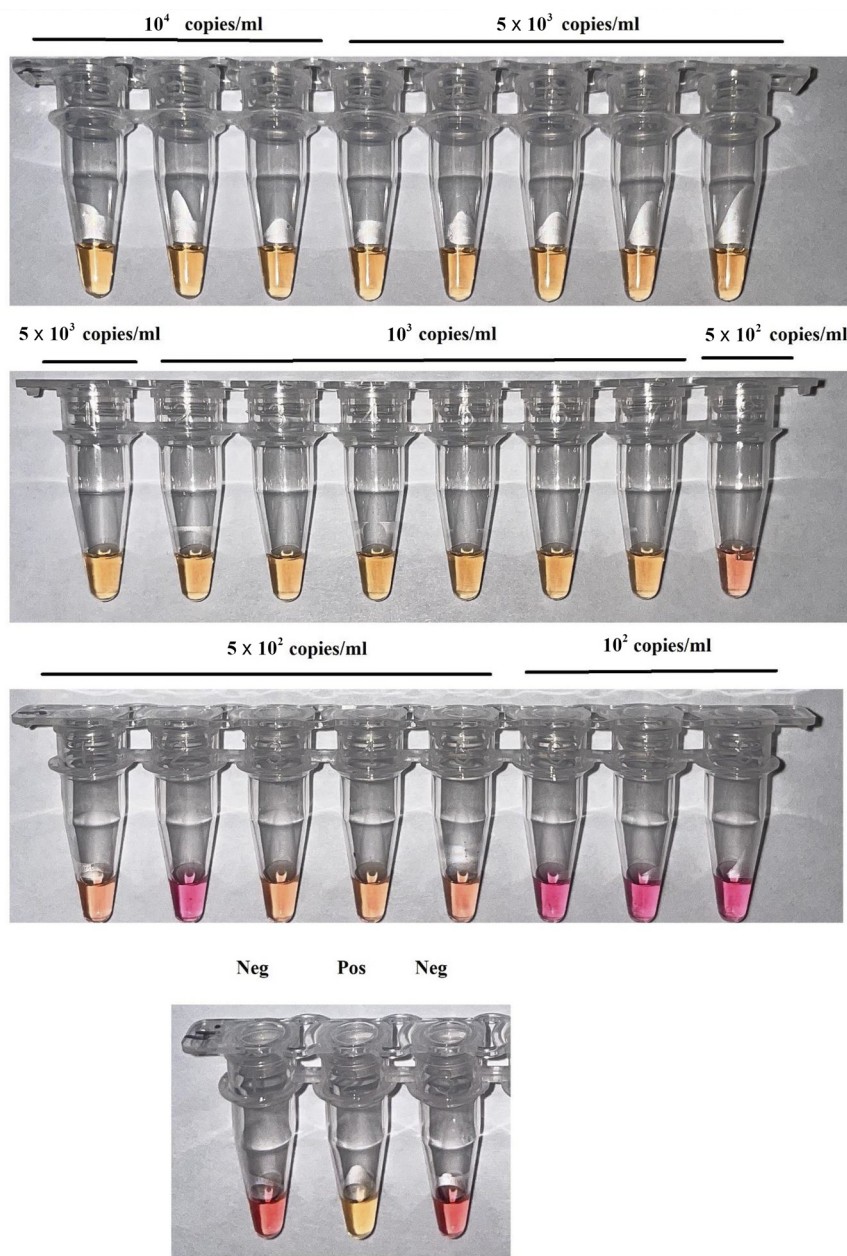

**FIG 1** Analytical sensitivity of the colorimetric c-LAMP. Visual detection with cresol red (pink = negative and yellow = positive). Serial dilutions of pAdvF plasmid DNA with concentrations of $10^4$ copies/mL (three replicates), $5 \times 10^3$ copies/mL (six replicates), $10^3$ copies/mL (six replicates), $5 \times 10^2$ copies/mL (six replicates), and $10^2$ copies/mL (three replicates) were used. Neg – negative control amplification without DNA; Pos – positive control amplification, $10^5$ copies/mL.

The specificity of LAMP for hAdV40/41 adenoviruses was confirmed on the DNA of related mastadenoviruses (see Table 5) and bacteria that cause intestinal infections. To determine the specificity of colorimetric c-LAMP and f-LAMP with assimilating FRET primer probe, the reaction time was increased to 60 min for both. No cross-reactions with other organisms were detected (Fig. 3 and 4).

**Diagnostic performance characteristics of c-LAMP and f-LAMP assays.** The clinical sensitivity and specificity of c-LAMP and f-LAMP assays were estimated using the clinical panel of 107 fecal extract samples consisting of 51 PCR-positive adenovirus F samples and 56 PCR-negative samples. The clinical samples were previously tested using the AmpliSens OKI-screen-FL kit (Table 1).

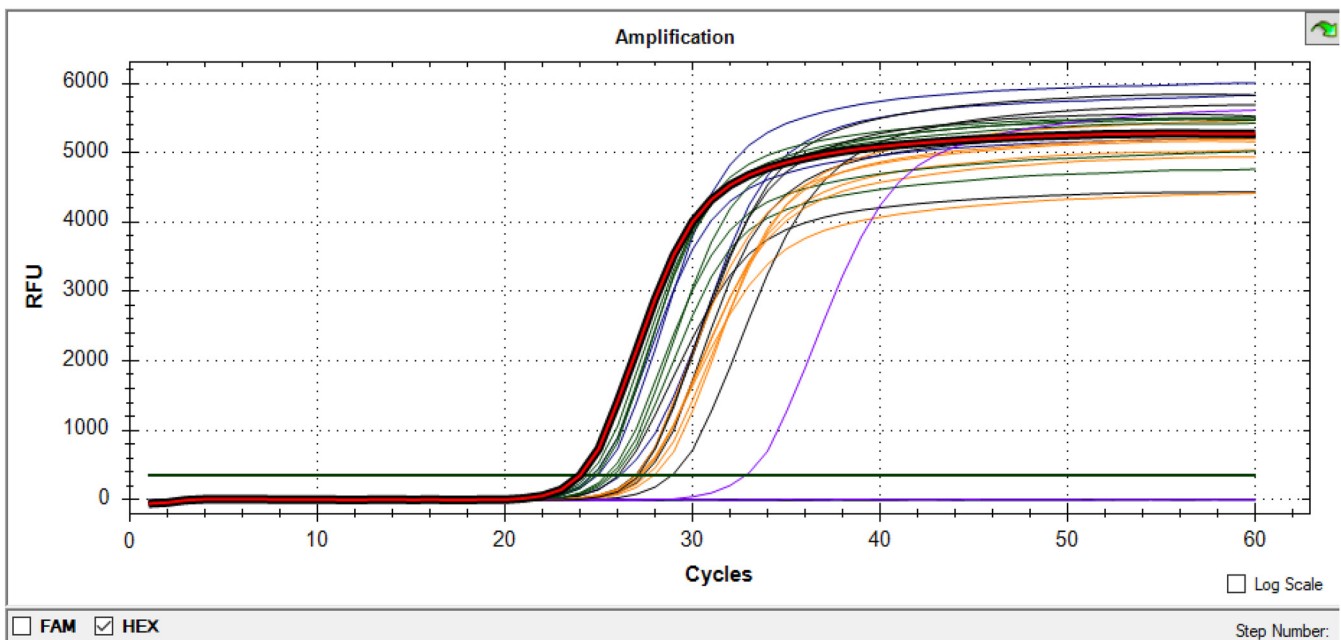

| sample | Ct*/2, min |
|---|---|
| 10^4 copies/ml | 13,1 |
| 10^4 copies/ml | 12,4 |
| 10^4 copies/ml | 12,3 |
| 5*10^3 copies/ml | 12,8 |
| 5*10^3 copies/ml | 12,7 |
| 5*10^3 copies/ml | 12,2 |
| 5*10^3 copies/ml | 12,3 |
| 5*10^3 copies/ml | 12,4 |
| 5*10^3 copies/ml | 12,9 |
| 10^3 copies/ml | 13,6 |
| 10^3 copies/ml | 13,8 |
| 10^3 copies/ml | 13,5 |
| 10^3 copies/ml | 13,6 |
| 10^3 copies/ml | 13,9 |
| 10^3 copies/ml | 13,5 |
| 5*10^2 copies/ml | 13,5 |
| 5*10^2 copies/ml | 13,0 |
| 5*10^2 copies/ml | 13,7 |
| 5*10^2 copies/ml | 13,5 |
| 5*10^2 copies/ml | 14,4 |
| 5*10^2 copies/ml | N/A |
| 10^2 copies/ml | N/A |
| 10^2 copies/ml | 16,4 |
| 10^2 copies/ml | N/A |
| Pos | 12,0 |
| Neg | N/A |
| Neg | N/A |

\* Ct - the cycle threshold value

**FIG 2** Analytical sensitivity of f-LAMP with assimilating FRET primer probe. Serial dilutions of pAdvF plasmid DNA with concentrations of $10^4$ copies/mL (three replicates), $5 \times 10^3$ copies/mL (six replicates), $10^3$ copies/mL (six replicates), $5 \times 10^2$ copies/mL (six replicates), and $10^2$ copies/mL (three replicates) were used. Pos, positive control amplification, $10^5$ copies/mL; Neg – negative control amplification without DNA; $C_T$, cycle threshold value.

Among the 51 PCR-positive adenovirus F samples, 31 samples had a sufficient viral load with PCR threshold cycles ($C_T$) of less than 36, and the 20 PCR-positive adenovirus F samples had threshold cycles of more than 36, which corresponds to a very low virus content in these samples.

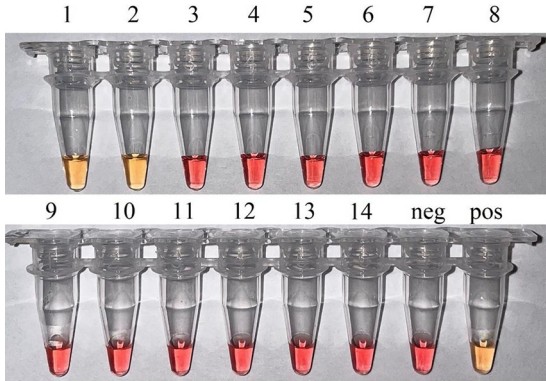

FIG 3 Analytical specificity of the colorimetric c-LAMP. Visual detection with cresol red (pink = negative and yellow = positive). Samples are described as follows: (1, 2) Isolates of hAdV40 and hAdV41 from fecal extract confirmed by Sanger sequencing of the hexon gene; (3) hAdV7; (4) hAdV21; (5) hAdV1; (6) hAdV2; (7) hAdV3; (8) hAdV4; (9) hAdV5; (10) hAdV6; (11) hAdV 8; (12) *Shigella flexneri* and *Shigella sonnei*; (13) *Salmonella enteritidis*; (14) hAdV40/41 negative fecal extract. Neg – negative control amplification without DNA; Pos – positive control amplification, $10^5$ copies/mL.

All 31 PCR-positive samples with a $C_T$ PCR of less than 36 were f-LAMP positive and c-LAMP positive. At a high adenoviral load (PCR $C_T \leq 15$; DNA > $10^8$ copies/mL), the time threshold of f-LAMP was 10 min, and the remaining samples with a PCR threshold of $15 \leq C_T \leq 36$ had a time threshold of f-LAMP of no more than 30 min. For colorimetric c-LAMP with visual control, the amplification time was also no more than 30 min, and samples with a high viral load (PCR $C_T \leq 15$) changed color in 10 to 15 min.

Among samples with a low adenoviral load (PCR $C_T > 36$), f-LAMP revealed adenovirus DNA in 12 of 20 PCR-positive samples, while c-LAMP revealed adenovirus DNA in only 4 (of 20) PCR-positive samples.

All 56 PCR-negative samples were determined to be negative by both f-LAMP and c-LAMP, even when the reaction time was increased to 60 min, demonstrating the high specificity of the developed LAMP assays.

The diagnostic sensitivity of f-LAMP with an assimilating probe was 86.3%, slightly inferior to the PCR sensitivity for samples with extremely low adenovirus load (Cohen's $\kappa$, 0.87).

PCR and colorimetric LAMP showed only a moderate agreement (Cohen's $\kappa$, 0.70), as c-LAMP was significantly inferior to PCR for samples with low adenovirus load; the sensitivity of c-LAMP was 68.7% (Table 2).

**Automated DNA extraction system versus manual method.** The manual DNA extraction method based on guanidine isothiocyanate lysis followed by alcohol DNA precipitation and an automated DNA extraction protocol based on DNA isolation on magnetic spheres were compared.

The potential for cross-contamination during automated DNA extraction was estimated using negative samples in random wells of the plate in close proximity to samples with a high adenoviral load. No cross-contamination was detected.

The automated extraction protocol was more effective than the manual protocol, and adenovirus DNA was detected in 53 of 107 samples instead of 51 PCR-positive samples according to the manual protocol (Table 3).

For DNA extracted by samples, a shift to earlier $C_T$ values was observed during the amplification of both adenovirus DNA ($\Delta C_T$av, 2.2) and DNA from exogenous internal control (IC) ($\Delta C_T$av, 1.8) added before extraction to the tested clinical material with automatic DNA isolation from the clarified fecal extract. In some cases, the value of $\Delta C_T$av for DNA isolated by the standard protocol and the automatic isolation protocol reached 7 cycles. Two discordant samples for which adenovirus DNA was not detected using the AmpliTest RiboPrep kit were positive for automatic DNA isolation ($C_T = 30.8$; $C_T = 38.1$).

**Incidence of mixed infections among children with acute adenovirus gastroenteritis.** Samples positive for adenovirus DNA ($n = 51$) were tested using the AmpliSens OKI-screen-FL PCR kit for the presence of various viral and bacterial pathogens.

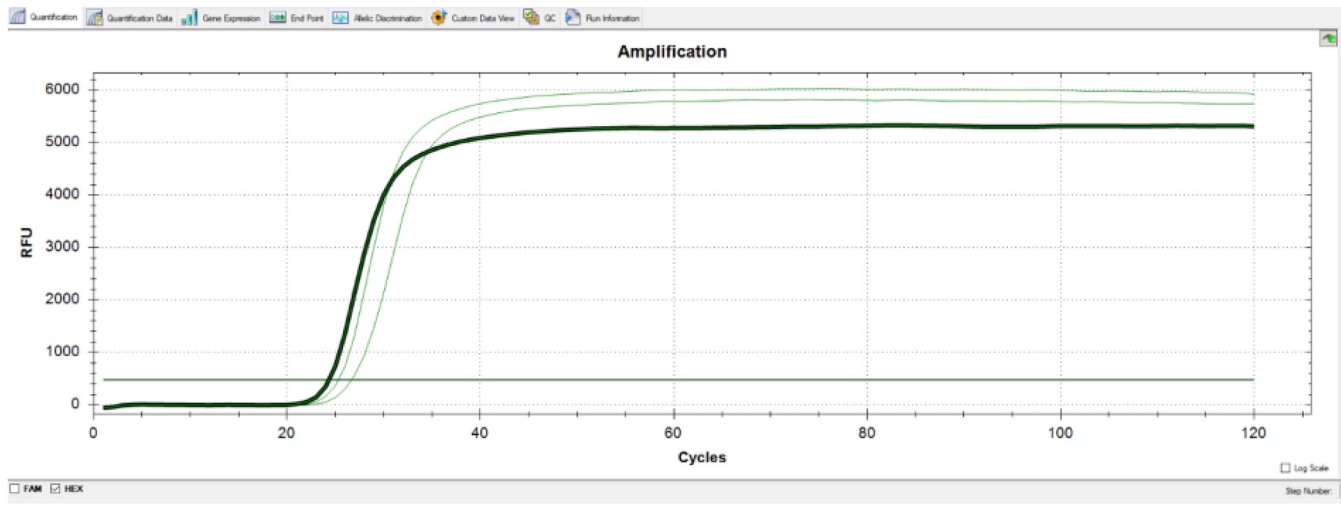

| Sample | Ct / 2, min |
|---|---|
| hAdV 40 | 12,7 |
| hAdV 41 | 13,3 |
| hAdV 7 | N/A |
| hAdV 21 | N/A |
| hAdV 1 | N/A |
| hAdV 2 | N/A |
| hAdV 3 | N/A |
| hAdV 4 | N/A |
| hAdV 5 | N/A |
| hAdV 6 | N/A |
| hAdV 8 | N/A |
| Shigella flexneri | N/A |
| Shigella sonnei | N/A |
| Salmonella enteritidis | N/A |
| *hAdV40/41* negative fecal extract | N/A |
| pos | 12,2 |
| neg | N/A |

**FIG 4** Analytical specificity of f-LAMP with assimilating FRET primer probe. Samples included the following: isolates of *hAdV40* and *hAdV41* from fecal extract confirmed by Sanger sequencing of the *hexon* gene; strains of *hAdV7, hAdV21, hAdV1, hAdV2, hAdV3, hAdV4, hAdV5, hAdV6, hAdV8, Shigella flexneri, Shigella sonnei,* and *Salmonella enteritidis*; *hAdV40/41* negative fecal extract. Pos – positive control amplification, $10^5$ copies/mL; Neg – negative control amplification without DNA.

In 2 cases, adenovirus F developed as a monoinfection, while in 49 cases, the infection was caused by various combinations of infectious agents as follows: rotavirus A/norovirus II/adenovirus F (20), rotavirus A/adenovirus F (18), rotavirus A/adenovirus F/*Campylobacter* (6), norovirus II/adenovirus F (2), rotavirus A/astrovirus/adenovirus F (1), rotavirus A/norovirus II/adenovirus F/*Campylobacter* (1), and rotavirus A/norovirus II/*Shigella*-enteroinvasive *Escherichia coli* (EIEC) (1) (Fig. 5).

## DISCUSSION

c-LAMP is fast and inexpensive, and it can be evaluated without any equipment. In the present study, the c-LAMP assay had good sensitivity for samples up to a PCR $C_T$ of ~36 with a turnaround of less than 30 min. Some false-negative results were obtained for samples with low viral load (PCR $C_T > 36$). However, the viral load in clinical specimens often reaches high concentrations, up to $1.5 \times 10^8$ genomic copies per reaction during acute adenovirus infection (23). Colorimetric LAMP allows visual detection in a short time, so this method is suitable even for budget-restricted laboratories in the absence of expensive PCR machines.

**TABLE 1** Comparison of PCR, c-LAMP, and f-LAMP results

| Sample | PCR AmpliSens OKI-screen-FL $C_T$ | f-LAMP FL $C_T$/2, min | c-LAMP, color change |
|---|---|---|---|
| 1 | 33.9 | 19.2 | + |
| 2 | 36.3 | 24 | N/D |
| 3 | N/D[a] | 19.4 | N/D |
| 4 | 37.3 | 22.9 | N/D |
| 5 | N/D | N/D | N/D |
| 6 | 31.3 | 17.7 | + |
| 7 | 9.1 | 8.3 | + |
| 8 | 26.5 | 17.8 | + |
| 9 | 28.1 | 18.5 | + |
| 10 | 27.4 | 15.7 | + |
| 11 | 13.0 | 9.3 | + |
| 12 | 32.5 | 17.7 | + |
| 13 | 33 | 19.1 | + |
| 14 | 36.4 | 19.8 | + |
| 15 | 34.5 | 20.0 | + |
| 16 | 42.3 | N/D | N/D |
| 17 | 34.0 | 22.1 | + |
| 18 | 31.3 | 17.2 | + |
| 19 | 30.7 | 18.0 | + |
| 20 | 13.4 | 9.0 | + |
| 21 | 30.0 | 16.5 | + |
| 22 | 36.6 | 24.2 | N/D |
| 23 | 31.3 | 16.9 | + |
| 24 | 35.3 | 22.9 | + |
| 25 | 29.2 | 15.7 | + |
| 26 | 40.7 | 25.8 | + |
| 27 | N/D | N/D | N/D |
| 28 | 13.3 | 9.2 | + |
| 29 | 37.8 | 20.5 | + |
| 30 | 33.1 | 17.1 | + |
| 31 | 36.2 | N/D | N/D |
| 32 | 34.3 | 20.8 | + |
| 33 | 38.8 | N/D | N/D |
| 34 | 32.3 | 18.0 | + |
| 35 | 33.6 | 18.8 | + |
| 36 | 32.4 | 18.2 | + |
| 37 | 32.0 | 18.1 | + |
| 38 | 34.5 | 18.8 | + |
| 39 | 32.5 | 19.0 | + |
| 40 | 36.4 | 26.5 | N/D |
| 41 | 29.7 | 16.4 | + |
| 42 | 30.9 | 17.0 | + |
| 43 | 42.0 | 29.1 | N/D |
| 44 | N/D | N/D | N/D |
| 45 | N/D | N/D | N/D |
| 46 | N/D | N/D | N/D |
| 47 | N/D | N/D | N/D |
| 48 | N/D | N/D | N/D |
| 49 | N/D | N/D | N/D |
| 50 | N/D | N/D | N/D |
| 51 | 41.7 | N/D | N/D |
| 52 | 39.4 | N/D | N/D |
| 53 | N/D | N/D | N/D |
| 54 | 41.8 | N/D | N/D |
| 55 | 42.23 | 26.3 | N/D |
| 56 | N/D | N/D | N/D |
| 57 | N/D | N/D | N/D |
| 58 | N/D | N/D | N/D |
| 59 | N/D | N/D | N/D |
| 60 | N/D | N/D | N/D |
| 61 | N/D | N/D | N/D |
| 62 | N/D | N/D | N/D |

**TABLE 1** (Continued)

| Sample | PCR AmpliSens OKI-screen-FL $C_T$ | f-LAMP FL $C_T/2$, min | c-LAMP, color change |
|---|---|---|---|
| 63 | N/D | N/D | N/D |
| 64 | N/D | N/D | N/D |
| 65 | 39.4 | 21 | N/D |
| 66 | N/D | N/D | N/D |
| 67 | 37.1 | 21.2 | N/D |
| 68 | N/D | N/D | N/D |
| 69 | N/D | N/D | N/D |
| 70 | 40.7 | N/D | N/D |
| 71 | N/D | N/D | N/D |
| 72 | 29.1 | 16.9 | + |
| 73 | N/D | N/D | N/D |
| 74 | N/D | N/D | N/D |
| 75 | N/D | N/D | N/D |
| 76 | N/D | N/D | N/D |
| 77 | N/D | N/D | N/D |
| 78 | N/D | N/D | N/D |
| 79 | N/D | N/D | N/D |
| 80 | N/D | N/D | N/D |
| 81 | N/D | N/D | N/D |
| 82 | N/D | N/D | N/D |
| 83 | 9 | 9.1 | + |
| 84 | 37.5 | N/D | N/D |
| 85 | N/D | N/D | N/D |
| 86 | 36.5 | 20.8 | + |
| 87 | N/D | N/D | N/D |
| 88 | N/D | N/D | N/D |
| 89 | N/D | N/D | N/D |
| 90 | N/D | N/D | N/D |
| 91 | N/D | N/D | N/D |
| 92 | N/D | N/D | N/D |
| 93 | N/D | N/D | N/D |
| 94 | N/D | N/D | N/D |
| 95 | N/D | N/D | N/D |
| 96 | N/D | N/D | N/D |
| 97 | N/D | N/D | N/D |
| 98 | N/D | N/D | N/D |
| 99 | N/D | N/D | N/D |
| 100 | N/D | N/D | N/D |
| 101 | N/D | N/D | N/D |
| 102 | N/D | N/D | N/D |
| 103 | N/D | N/D | N/D |
| 104 | N/D | N/D | N/D |
| 105 | N/D | N/D | N/D |
| 106 | N/D | N/D | N/D |
| 107 | N/D | N/D | N/D |

[a]N/D, not detected.

The sensitivity of the f-LAMP assay using fluorescently labeled assimilating primer probes was not significantly lower than that of PCR, but the turnaround took less than 30 min. Acute gastroenteritis is often caused by various combinations of mixed infections. Such mixed infections can aggravate a disease; for example, the presence of histo-blood group antigen (HBGA)-expressing bacteria promotes better binding of viral particles to cells and increases replication efficiency (24) and thermal stability (25), decreasing the effectiveness of specific immunity (26). Viral coinfection occurs in 29% cases of acute gastroenteritis and causes a more severe course (27).

Coinfected patients had lower base excess, which indicated metabolic acidosis as a result of increasing dehydration and severe wasting (28).

Adenovirus infection is often observed in combination with other viruses. The most common combinations are adenovirus/norovirus and rotavirus/norovirus, and these infections accompany adenovirus in 19% and 34% of cases, respectively (29).

**TABLE 2** Diagnostic performance characteristics of LAMP

| Compared to the results from PCR | f-LAMP FL[a] | | c-LAMP[b] | |
|---|---|---|---|---|
| | Value | Interval, $P = 95\%$ (%) | Value | Interval, $P = 95\%$ (%) |
| Specificity | 100 | 93.6–100.0 | 100 | 93.6–100.0 |
| Sensitivity | 86.3 | 73.8–94.3 | 68.7 | 54.1–80.9 |
| Cohen's kappa | 0.87 | | 0.70 | |

[a]For f-LAMP, 44 (51) positive and 56 (56) negative samples were correctly identified.
[b]For c-LAMP, 35 (51) positive and 56 (56) negative samples were correctly identified.

Therefore, the development of rapid tests in the multiplex f-LAMP format is a perspective direction, especially in the case of acute gastroenteritis diagnostics. In addition, f-LAMP may use the amplification reaction of the internal control sample to assess the efficiency of the extraction of nucleic acids.

DNA extraction is the primary and most important step in molecular diagnosis, so strong consideration must be paid to select a method of nucleic acid extraction. Clinical samples contain many biologically active substances (nucleases, enzyme inhibitors) that can influence the results of the study; hence, the choice of the DNA extraction method is one of the most important stages in development. DNA extraction methods should be easy to use and not expensive; they should also be efficient and reproducible. The success of the subsequent analysis depends on the isolation of DNA fragments that have good purity, integrity, and concentration.

In clinical practice, one of the most common methods of nucleic acid extraction is the use of chaotropic agents, such as guanidine (30, 31). Guanidine isothiocyanate denatures proteins, including nucleases and peptidoglycans, and destroys cells (32). Guanidine salts at low concentrations do not inhibit amplification; moreover, according to some reports, their efficiency can be increased (33). Protocols based on cell lysis and protein denaturation in a solution of guanidine isothiocyanate followed by alcohol precipitation of nucleic acids are not laborious and make it possible to obtain nucleic acids suitable for amplification rather quickly.

Using separation technology on magnetic spheres can significantly reduce the analysis time, especially in the case of a large flow of investigated samples, and can be applied to automate the extraction of DNA/RNA from various clinical materials, including swabs (34), sputum (35), blood plasma (36), and feces (37).

After analyzing many clinical samples ($n = 107$), it was shown in this study that the automatic isolation protocol shows a decrease in the values of the threshold amplification cycles for both DNA and exogenous IC compared with those of the manual extraction protocol of the AmpliTest RiboPrep kit, which is in widespread use clinical trials in the Russian Federation. This may be related to a higher extraction efficiency and/or a more efficient removal of inhibitors during sorption on a magnetic sorbent. Thus, the most important trends in the development of modern methods of molecular diagnostics are their acceleration, automation, and development of tests in the "patient's bedside" format in combination with maintaining and increasing their sensitivity and specificity.

In the near future, the application of fast and accurate tests based on isothermal amplification will provide timely results regarding the pathogens of acute gastroenteritis, promoting early determination of treatment and recovery tactics.

**TABLE 3** Results of DNA extraction from clinical samples using manual and automated DNA extraction protocols

| Protocol | AmpliSens OKI screen-FL adenovirus F DNA | |
|---|---|---|
| | PCR positive | PCR negative |
| Automated viral DNA/RNA extraction kit, GeneRotex 96 | 53 | 54 |
| Manual AmpliTest RiboPrep | 51 | 56 |

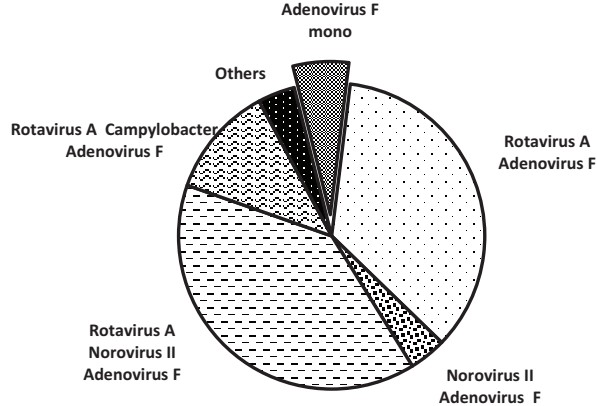

| samples | Adenovirus F PCR positive samples, n=51 | | | | | | |
|---|---|---|---|---|---|---|---|
| | Mono | *Rotavirus A* | *Norovirus II* | *Campylobacter* | *Shigella + EIEC* | *Astrovirus* | *Salmonella* |
| n | 2(51) | 47(51) | 25(51) | 7(51) | 1(51) | 1(51) | 1 (51) |
| % | 4.0 | 92.2 | 49.0 | 13.7 | 2.0 | 2.0 | 2.0 |

**FIG 5** Incidence of mixed infections among children with acute adenovirus gastroenteritis. Rotavirus A was detected in the vast majority of cases (92.2%) of adenovirus infection, and norovirus II and campylobacteriosis were detected in 49.0% and 13.7% of cases, respectively.

## MATERIALS AND METHODS

**Clinical samples.** Fecal samples (*n* = 107) from children with symptoms of acute gastroenteritis were obtained from Children's City Clinical Hospital No. 9, named after G.N. Speransky of the Moscow Health Care Department (Moscow, Russia). Ethical approval for using clinical samples for research purposes was obtained from the local ethics committee at Children's City Clinical Hospital No. 9, Moscow, Russia (protocol no. 44, 19.04.22).

To prepare the fecal extract, 0.1 g (0.1 mL) of feces was resuspended in 0.8 mL of phosphate salt buffer (VWR International, LLC, USA). The homogeneous suspension was centrifuged at $10,000 \times g$ on a MiniSpin (Eppendorf, Germany) for 5 min, and the supernatant was collected and stored at −70°C before use.

**Extraction of DNA.** For DNA extraction in manual mode, an AmpliTest RiboPrep kit (FSBI "CSP" FMBA, Russia) was used for extraction DNA. The method is based on guanidinium thiocyanate lysis followed by the isopropanol precipitation of nucleic acids and dissolution of the pellet in the elution buffer (10 mM Tris-HCl, 1 mM EDTA [pH 8.0]).

For DNA extraction in automatic mode using magnetic particle techniques, a viral DNA and RNA extraction kit (Xi'an Tianlong Science and Technology Co., LTD, China) and a Tianlong GeneRotex 96 nuclear acid extraction system (Xi'an Tianlong Science and Technology Co., LTD, China) were used.

Exogenous internal control (IC) from the AmpliSens OKI-screen-FL kit (Central Research Institute of Epidemiology, Russia) was added to each sample. DNA was extracted from 100 $\mu$L of the fecal extract and eluted in 100 $\mu$L of elution buffer for both modes of extraction.

**TABLE 4** LAMP primers for the detection of hAdV40/41 hexon gene

| Oligonucleotide | Name | Sequence (5′–3′)[a] | Length (nt) |
|---|---|---|---|
| Forward outer primer | AdvF3.1 | TGTATGCGCCTCCTGTGTTA | 20 |
| Forward outer primer | AdvF3.2 | TGTGTACGCCTCCTGTGTTA | 20 |
| Reverse outer primer | AdvB3 | ACRAAKCGCAGCGTCAGTC | 19 |
| Loop primer | AdvFL | GCATGTAAGACCATTGCGGCATCA | 24 |
| Loop primer | AdvBL | ACSTACTTCAGCCTGGGGAACAA | 23 |
| Forward inner primer | AdvFIP | ACCAGGCCCGGRCTCAGRTATTTTGCCAGASAGCCGAGTGAC | 42 |
| Reverse inner primer | AdvBIP1 | CARTTTGCCCGCGCCACCGATTTTTTACATCGTGGGTSGGAGC | 43 |
| Reverse inner primer | AdvBIP2 | CAGTTCGCCCGTGCCACCGATTTTTTACATCGTGGGTSGGAGC | 43 |
| Probe/loop primer | AdvFL.NatTail | R6G NatTail-GCATGTAAGACCATTGCGGCATCA[b] | |
| Quencher | FLQ | Common quencher -BHQ1[b] | |

[a]Degenerate nucleotides: S = G or C, K = G or T, R =A or G.
[b]The nucleotide sequences of *NatTail* and the complementary strand *Common quencher* with BHQ1 are from reference 40.

**TABLE 5** Human mastadenovirus strains used in this work

| Human mastadenovirus | hAdV type | Titer (log TCID 50/mL) |
|---|---|---|
| B | 7 | 4.5 |
| | 21 | 3.0 |
| C | 1 | 3.0 |
| | 2 | 3.0 |
| | 3 | 3.0 |
| | 4 | 5.0 |
| | 5 | 5.0 |
| | 6 | 5.0 |
| D | 8 | 3.0 |

**PCR.** The samples of the clinical panel (107 fecal extracts) were PCR tested for the presence of adenovirus F. Real-time PCR was performed using an AmpliSens OKI-screen-FL kit (Central Research Institute of Epidemiology, Russia) in CFX96 real-time PCR machines (Bio-Rad, USA).

AmpliSens OKI-screen-FL kit specifically detects adenovirus F and does not detect other types of adenoviruses. Declared analytical sensitivity for adenovirus F DNA isolated from fecal extract is up to $10^4$ copies/mL.

Adenovirus F-positive samples were selected and tested by AmpliSens OKI-screen-FL kit for the presence of other enteric viral and bacterial pathogens (rotavirus A, norovirus 2, astrovirus, *Shigella* spp., *E. coli* [EIEC], *Salmonella* spp., and *Campylobacter* spp.) included in the OKI-screen-FL test.

**Design of LAMP primers.** Multiple sequence alignments of hAdV sequences were performed using the Clustal W algorithm in Mega X (38). LAMP target-specific primers specific for hAdV40/41 were designed using Primer Explorer V (Fujitsu, Japan) on unique template regions of hAdV40/41 hexon genes that are not similar to hexon genes of human mastadenovirus A, B, C, D, G; bat mastadenovirus; canine mastadenovirus; deer mastadenovirus; dolphin mastadenovirus; murine mastadenovirus; ovine mastadenovirus; porcine mastadenovirus; simian mastadenovirus; and other taxons.

Using the search tool NCBI BLAST (39), it was shown the primer sequences do not have high identity to other taxon genomes from GenBank.

The single-nucleotide polymorphisms in the LAMP primers among hAdV40/41 adenovirus sequences were accounted for using degenerate nucleotides.

Primer sequences are listed in Table 4. All oligonucleotides were synthesized and high-pressure liquid chromatography (HPLC) purified (JSC Gentera, Russia).

**Colorimetric LAMP reaction.** Colorimetric LAMP reactions were performed using WarmStart Colorimetric LAMP 2× master mix (DNA and RNA) (New England Biolabs, USA) using 2.5 $\mu$L of 10× LAMP primer mix with appropriate concentrations—AdvFIP (16 $\mu$M), AdvBIP1 (8 $\mu$M), AdvBIP2 (8 $\mu$M), AdvB3 (8 $\mu$M), AdvFL (8 $\mu$M), AdvBL (8 $\mu$M), AdvF3.1 (2 $\mu$M), and AdvF3.2 (2 $\mu$M)—added to 12.5 $\mu$L of the above and 10 $\mu$L of sample for a total reaction volume of 25 $\mu$L. The total incubation time at 65°C was ~40 min.

The sample was considered negative if the pink color of the phenolic red remained at the end of the reaction and positive if the color of the phenolic red changed to orange-yellow.

Positive and negative controls were used in each experiment.

The results were independently visually evaluated by two persons, and photographs were taken.

**Real-time LAMP with assimilating FRET primer-probe.** For real-time LAMP, fluorescently labeled assimilating primer probes with complementary quenching-labeled oligonucleotide strands were used. The FRET assimilating AdvFL.NatTail/FLQ primer-probe (Table 4) consists of two partially complementary oligonucleotides. One of these oligonucleotides is an AdvFL loop primer with an R6G fluorescently labeled NatTail tag at the 5′ end and a second oligonucleotide. FLQ is complementary to the NatTail sequence common quencher labeled at the 3′ end by a BHQ1 quencher. The sequences of the NatTail tag and the complementary oligonucleotide common quencher were described early in reference 40. Annealing of the 3′ overhang of the fluorescent strand AdvFL.NatTail to the loop region of the LAMP product initiates a new polymerization reaction, which ultimately results in displacement of the quencher strand FLQ and increases the fluorescent signal.

Real-time LAMP reactions were performed using 8 U of Bst polymerase and 5× LAMP buffer (340 mM Tris-HCl, pH 8.8 [Sigma-Aldrich, USA], 40 mM MgSO$_4$ [New England Biolabs, USA], 100 mM (NH$_4$)$_2$SO$_4$ [Sigma-Aldrich, USA], and 0.2 mM bovine serum albumin [BSA] [Genterra, Russia]), 1.4 mM each deoxynucleoside triphosphate (dNTP) (Biosan, Russia), and 10× LAMP primer mix with appropriate concentrations as follows: AdvFIP (16 $\mu$M), AdvBIP1 (8 $\mu$M), AdvBIP2 (8 $\mu$M), AdvB3 (8 $\mu$M), AdvFL (8 $\mu$M), AdvBL (4 $\mu$M), AdvF3.1 (2 $\mu$M), AdvF3.2 (2 $\mu$M), primer-probe AdvFL.NatTail (0.8 $\mu$M)/FLQ (1.6 $\mu$M), and 10 $\mu$L of sample per reaction. The reaction volume was 25 $\mu$L. CFX96 real-time PCR machines (Bio-Rad, USA) were used, and the reaction time at 65°C was ~40 min, with fluorescence determination every 30 s.

Positive and negative controls were used in each experiment.

**Positive control pAL2-AdvFpAL2-AdvF for LAMP assays.** Amplification of the hexon gene fragment was performed in 25-$\mu$L reaction volumes containing 0.8 $\mu$M AdvB3, 0.4 $\mu$M AdvF3.1, 0.4 $\mu$M AdvF3.2, 0.25 mM each dNTP (Biosan, Russia), 5 U TaqF polymerase, 70 mM Tris-HCl pH 8.3 (Sigma-Aldrich, USA), 4 mM MgCl$_2$ (Sigma-Aldrich, USA), 80 mM KCl (Sigma-Aldrich, USA), 0.2 mM BSA (Genterra, Russia), and 10 $\mu$L of template DNA isolated from clinical species. CFX96 real-time PCR machines (Bio-

Rad, USA) were used, and the cycling conditions were 95°C for 15 min, 40 cycles of 95°C for 15 s, 60°C for 30 s, and 72°C for 15 s.

The PCR product was inserted into the linear plasmid vector pAL2-TA (Evrogen, Russia) with T sticky ends.

The sequence of the cloned fragment of the hexon gene was confirmed by Sanger sequencing and placed in GenBank under the accession number MW570718.1.

The concentration of plasmid DNA was evaluated using digital PCR on a QX200 Droplet Digital PCR system (Bio-Rad Laboratories, USA) with a mixture of Eva Green Super mix (Bio-Rad Laboratories, USA) and primers AdvF3.1 and AdvF3.2 at 0.125 mM and AdvB3 at 0.25 mM. The thermal cycling program was 95°C for 5 min followed by 40 cycles at 94°C for 30 s, 60°C for 60 s, 4°C for 5 min, and 90°C for 5 min.

**Evaluating the sensitivity and specificity of colorimetric LAMP and LAMP in real time with an assimilating probe.** The sensitivity of the LAMP methods was evaluated using 10-fold serial dilutions of the plasmid pAdvF, ranging from $10^2$ to $10^4$ copies/mL (concentration of template input), in three repeats for $10^2$-copy/mL and $10^4$-copy/mL dilutions and in six repeats at concentrations of $5 \times 10^2$ copies/mL, $10^3$ copies/mL, and $5 \times 10^3$ copies/mL close to the cutoff value of LAMP performance for each dilution.

It was accepted that the analytical sensitivity of the assay was the lowest quantity of a pAdvF that was determined to be positive for three repeats.

The specificity of the LAMP assays (no cross-reactivity with other pathogens) was evaluated using strains of human mastadenovirus (Table 5) provided by Smorodintsev Influenza Research Institute and the enterobacteria *Shigella flexneri* ($1 \times 10^6$ CFU/mL), *Shigella sonnei* ($1 \times 10^6$ CFU/mL), and *Salmonella enteritidis* ($1 \times 10^6$ CFU/mL) provided by our own collection of FSBI "CSP" of FMBA and hAdV40/41-negative fecal extract.

DNA was extracted from 100 $\mu$L of stock cultures of viral and microbial strains and eluted in 100 $\mu$L of elution buffer using the AmpliTest RiboPrep kit (FSBI "CSP" FMBA, Russia).

**Statistical data analysis.** Diagnostic sensitivity and specificity were calculated by MEDCALC (https://www.medcalc.org/calc/diagnostic_test.php) using information about true (TP) and false (FP) positive and true (TN) and false (FN) negative results of LAMP assays.

To measure the degree of agreement between PCR and LAMP assays, Cohen's kappa was used. Cohen's kappa index was estimated as weak if below 0.40, moderate if between 0.41 and 0.60, substantial if between 0.61 and 0.80, and almost perfect between 0.81 and 1.00 (41).

**Data availability.** The GenBank/EMBL/DDBJ accession numbers for the target fragment of the hexon gene are MW570718.1.

## ACKNOWLEDGMENT

This work was supported by Federal Medical Biological Agency of Russia.

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
