## [Reviewer comments · Microbiology Spectrum]

Microbiology Spectrum

Development and application of LAMP assays for the detection of enteric adenoviruses in feces.

Anna Shuryaeva, Tatyana Malova, Anna Tolokonceva, Sofia Karseka, Maria Gordukova, Ekaterina Davydova, and German Shipulin

Corresponding Author(s): Ekaterina Davydova, Federal State Budgetary Institution Centre for Strategic Planning and Management of Biomedical Health Risks of the Federal Medical Biological Agency

Review Timeline:

Submission Date:	February 15, 2022
Editorial Decision:	March 8, 2022
Revision Received:	May 6, 2022
Editorial Decision:	May 20, 2022
Revision Received:	June 6, 2022
Accepted:	June 7, 2022

Editor: Karen Carroll

Reviewer(s): Disclosure of reviewer identity is with reference to reviewer comments included in decision letter(s). The following individuals involved in review of your submission have agreed to reveal their identity: Matthew Martin Hernandez (Reviewer #1); Jeremy Ratcliff (Reviewer #3)

Transaction Report:

DOI: <https://doi.org/10.1128/spectrum.00516-22>

March 8, 2022

Dr. Ekaterina Evgenevna Davydova
Federal State Budgetary Institution Centre for Strategic Planning and Management of Biomedical Health Risks of the Federal
Medical Biological Agency
Pogodinskaya Str, 10
Moscow 119121
Russia

Re: Spectrum00516-22 (Development and application of LAMP assays for the detection of enteric adenoviruses in feces.)

Dear Dr. Ekaterina Evgenevna Davydova:

Thank you for submitting your manuscript to Microbiology Spectrum. Your paper has been thoroughly reviewed by 3 experts in molecular virology diagnostics. The reviewers note a lack of clarity regarding methods and study design and have provided many recommendations to improve the manuscript. Please respond to the questions regarding ethical approval directly in the manuscript as well.

Link Not Available

Sincerely,

Karen Carroll

Journals Department
Reviewer comments:

Reviewer #1 (Comments for the Author):

Sharuyeva et al. have developed a colorimetric and fluorescent LAMP assays (cLAMP and fLAMP) to detect human adenovirus F (hAdV40 and hAdV41) in pediatric patients presenting with acute gastroenteritis. They utilize pediatric clinical stool specimens to test for adenovirus F viruses by their novel methods in comparison to a real-time PCR (RT-PCR) assay. They conclude their LAMP assays have an analytic sensitivity comparable to that of RT-PCR and have the potential for scalability by incorporating

automated nucleic acid extraction. While these findings have promise for utilization in under-resourced areas, the data presented has a number of issues that warrant major revisions and re-analyses.

Reviewer #2 (Comments for the Author):

The manuscript by Malova describes a LAMP assay for enteric adenoviruses in feces of children. The manuscript is lacking in details in several aspects detailed below:

1. Abstract should reflect the targets in this assay (hexon gene of adenovirus).
2. Provide clarity regarding primer design. For example in Table 1. it would be useful to indicate which ones are probes. Also indicate which target is being detected.
3. How was specificity of primers and probes tested?
4. Table 5. How was discrepancy resolution performed for the 2 samples that were positive after automated extraction?
5. Both results and discussion section can be clarified by including definite details on what the targets were for each assay and what the comparator assay was. As currently written, it is unclear whether assay for all targets including rotavirus and norovirus were in this study or whether this was a commercial assay that the LAMP assay was being compared to?
6. A brief explanation on how this assay can be utilized clinically will be valuable. It is clear that specimens with low viral load may not be detected by this assay. Is there a way that the rapidity of the test can be utilized in clinical decision making that would still make the test useful despite its insensitivity? Especially since it is low cost?

Reviewer #3 (Comments for the Author):

Shuryaeva et al. report on the development of a LAMP assay for detecting adenovirus F DNA from patient clinical samples. The authors report high sensitivity and specificity. The authors demonstrate high performance on a panel of fecal samples collecting from children experiencing acute viral gastroenteritis. In several locations in the manuscript, the validation could use more description or development.

Major Comments:

- There is no mention of ethical approval for the study. Was patient consent received for using clinical samples for research purposes? Patient clinical features were also mentioned ("Coinfected patients had lower base excess").
- Figure 1, panel 10³ copies/mL appears to be manipulated and represents two separate experiments. Can the authors please clarify? If the samples with equal concentration were run on different strips that is fine but the images shouldn't be edited to look as if they are one continuous image (and I would appreciate seeing the raw, unedited images). If the authors have more than three replicates for each concentration, can that additional data be shown? Six versus three replicates would greatly improve the interpretability of the sensitivity experiments as only three samples is insufficient. In general, I strongly recommend the authors increase the numbers of replicates at concentrations close to the cut-off value of LAMP performance and perform probit regression to obtain 95% and 50% limit of detection estimates.
- The claim that cLAMP and fLAMP have comparative sensitivity with PCR, the authors should demonstrate the sensitivity of PCR targeting the same region. This is also hard to interpret along the section "Among the 51 PCR-positive adenovirus F samples, 31 samples had a sufficient viral load with Ct PCR threshold cycles of less than 36, which corresponds to a virus content of 10³ copies/ml or more. The 20 PCR-positive adenovirus F samples had threshold cycles of more than 36, which corresponds to a very low virus content in these samples". If the PCR test was producing positive results for a higher proportion of samples with a "virus content" less than 10³ copies/ml than cLAMP or fLAMP surely that means PCR is more sensitive?
- Please provide the quantification of other targets used when testing the assay specificity. Were these tested on both cLAMP and fLAMP?
- The phrase "Additionally, the f-LAMP assay was able to detect several pathogens in one multiplexed reaction." is unclear. Did the authors develop a multiplexed LAMP reaction? That would be a strong development and would increase the utility of the technical advancements presented in this article.

Minor Comments:

- Please define the acronyms LAMP, PCR, and Ct in the main text of the article.
- In the methods, the authors state the samples were tested on serial dilutions from 5*10² copies/ml to 10⁶ copies/ml. The results section states that cLAMP was tested from 10² copies/mL to 10⁴ copies/mL and the fLAMP studies only show to 10⁴ copies/mL. Please clarify.
- The inclusion of "Ct" values for fLAMP is confusing as the PCR machine isn't really "cycling". It may result in improper comparison with qPCR methods. I would remove the Ct column from column 2 and only leave the time column.

Staff Comments:

Preparing Revision Guidelines

Please return the manuscript within 60 days; if you cannot complete the modification within this time period, please contact me. If you do not wish to modify the manuscript and prefer to submit it to another journal, please notify me of your decision immediately so that the manuscript may be formally withdrawn from consideration by Microbiology Spectrum.

Summary

Sharuyaeva et al. have developed a colorimetric and fluorescent LAMP assays (cLAMP and fLAMP) to detect human adenovirus F (hAdV40 and hAdV41) in pediatric patients presenting with acute gastroenteritis. They utilize pediatric clinical stool specimens to test for adenovirus F viruses by their novel methods in comparison to a real-time PCR (RT-PCR) assay. They conclude their LAMP assays have an analytic sensitivity (e.g., limit of detection (LoD)) comparable to that of RT-PCR (1000 cp/mL) and have the potential for scalability by incorporating automated nucleic acid extraction. While these findings have promise for utilization in under-resourced areas, the data presented has a number of issues that warrant major revisions and re-analyses.

Major Issues

1. On p. 3, based on this study, the group is utilizing human specimens for research purposes. However, are these specimens residual, de-identified, discard specimens or are these specimens directly collected from children? Moreover, has this study received approval by an internal review board to ensure that children were appropriately consented in this study if the team interacted with them? If so, the team should please elaborate to ensure that this study is ethically sound.
2. On p. 6, it seems as if nucleic acids from viral and bacterial pathogens utilized for specificity testing was extracted by a different method (e.g., AmpliTest-RIBOT) compared to the adenovirus clinical specimens (e.g., RiboPrep). This may be a confounding variable to the assay and warrants discussion or evidence that nucleic acid extraction efficiency is consistent between methods.
3. The negative controls utilized in the assays are not optimal when determining analytical sensitivities of the assays. From a molecular perspective, in these studies, these are non-template controls as they do not include DNA (p. 7, Fig. 1 legend). Thus, negative controls warrant reactions using extracted DNA from healthy donors or pathogen-negative stool or even empty vector backbone (pAL2-TA). Please include the aforementioned template DNA in the cLAMP assay for Fig. 1. Furthermore, this is important to include in a repeat of fLAMP assay to appropriately inform cycle-threshold (Ct) cutoffs for calling presence/absence of adenoviral nucleic acids (see next point).
4. The Cts of replicates at the stated LoD concentration of the fLAMP assay (1000 cp/mL) are remarkably different (37-49, p. 8). This is hard to believe and should be re-evaluated with negative controls that utilize adeno-negative DNA as mentioned in #3 in order to define cutoffs for what defines a positive or negative result.
5. There are multiple issues with the diagnostic sensitivity/specificity analyses. In order to assess the diagnostic sensitivity/specificity of LAMP assays, you are comparing to the results from RT-PCR assay as the 'gold-standard' (p. 8). It is important to indicate what the cutoffs are for positive/negative specimens on the RT-PCR assay to inform how Ct values in the RT-PCR assay reflect viral genome quantity. Moreover, Table 3 and 4 are confusing / not informative. I would recommend making tables that demonstrate positive and negative results across the gold-standard versus novel method of comparison (e.g., cLAMP or fLAMP). Furthermore, agreement analyses are worthwhile (e.g., Cohen's kappa) to determine the utility of the LAMP assays (in addition to specificity and sensitivity statistical analyses. If still of interest, linear regressions of RT-PCR versus c/fLAMP Ct values can be performed to compare.
6. The comparison of automated versus manual DNA extraction methods for the gold-standard RT-PCR method (p. 10) is irrelevant for a study that compares novel LAMP diagnostic methods. If anything, extraction methods should be compared among each of the cLAMP and fLAMP methods.

Minor issues

1. For those unfamiliar with the RT-PCR assay utilized as the 'gold-standard', please indicate whether the assay can detect and differentiate between other adenovirus subtypes. Moreover, please indicate what cutoffs are used by the clinical laboratory (or in accordance with manufacturer's recommendations) to define positive or negative specimens as well as whether specimens are run as singletons or in replicates.

2. On p. 4-5, please indicate how the cLAMP assay is read out. Is this qualitatively read by eye or by a colorimeter? If by the former, it would be helpful to discuss the reliability of read-out by eye particularly by various users.

3. On p. 8, please elaborate what the Ct values were of the mastadenoviruses and bacteria tested in the specificity assay. This will depend on what your cutoff is for your defining positive/negative results.

4. Grammar and language:

- Italicize genus and species names of pathogens tested for specificity (e.g., p. 6).
- "Primer sensitivity" is misleading as BLAST-ing sequences does not inform sensitivity (e.g., lowest concentration of detection) of primers/target/assay (p. 7).
- Please go through the manuscript to ensure English grammar is intact/appropriate.

Reviewer comments:

Reviewer #1 (Comments for the Author):

Sharuyeva et al. have developed a colorimetric and fluorescent LAMP assays (cLAMP and fLAMP) to detect human adenovirus F (hAdV40 and hAdV41) in pediatric patients presenting with acute gastroenteritis. They utilize pediatric clinical stool specimens to test for adenovirus F viruses by their novel methods in comparison to a real-time PCR (RT-PCR) assay. They conclude their LAMP assays have an analytic sensitivity (e.g., limit of detection (LoD)) comparable to that of RT-PCR (1000 cp/mL) and have the potential for scalability by incorporating automated nucleic acid extraction. While these findings have promise for utilization in under-resourced areas, the data presented has a number of issues that warrant major revisions and re-analyses.

1-1. On p. 3, based on this study, the group is utilizing human specimens for research purposes. However, are these specimens residual, de-identified, discard specimens or are these specimens directly collected from children? Moreover, has this study received approval by an internal review board to ensure that children were appropriately consented in this study if the team interacted with them? If so, the team should please elaborate to ensure that this study is ethically sound.

Author's answer: Fecal samples were obtained from G.N. Speranskiy Children Hospital No 9, Moscow, Russian Federation in 2020-2021. All samples were collected for medical care in the hospital but the authorized representatives of children have signed an Informed Consent which allow to use sample for scientific research too. Thus, these specimens are residual, de-identified specimens and can be used in scientific research. Ethical approval for using these clinical samples for research purposes was obtained from the local ethics committee at Children's City Clinical Hospital No. 9, Moscow, Russia (protocol № 44, 19.04.22).

1-2. On p. 6, it seems as if nucleic acids from viral and bacterial pathogens utilized for specificity testing was extracted by a different method (e.g., AmpliTest-RIBOT) compared to the adenovirus clinical specimens (e.g., RiboPrep). This may be a confounding variable to the assay and warrants discussion or evidence that nucleic acid extraction efficiency is consistent between methods.

Author's answer Excuse me. AmpliTest-RIBOT is typing error. We use for manual nucleic acid extraction only AmpliTest RiboPrep kit (FSBI "CSP" FMBA, Russia). The corrections have been made.

1-3. The negative controls utilized in the assays are not optimal when determining analytical sensitivities of the assays. From a molecular perspective, in these studies, these are non-template controls as they do not include DNA (p. 7, Fig. 1 legend). Thus, negative controls warrant reactions using extracted DNA from healthy donors or pathogen-negative stool or even empty vector backbone (pAL2-TA). Please include the aforementioned template DNA in the cLAMP assay for Fig. 1. Furthermore, this is important to include in a repeat of fLAMP assay to appropriately inform cycle-threshold (Ct) cutoffs for calling presence/absence of adenoviral nucleic acids (see next point).

Author's answer We repeated the experiments to increase the repeats and using new sample - DNA isolated from adenovirus-negative stool

1-4 The Cts of replicates at the stated LoD concentration of the fLAMP assay (1000 cp/mL) are remarkably different (37-49, p. 8). This is hard to believe and should be re-evaluated with negative controls that utilize adeno-negative DNA as mentioned in #3 in order to define cutoffs for what defines a positive or negative result.

Author's answer *We believe that the non-reproducibility of LAMP Cts was observed because the previously used LAMP buffer (5X LAMP buffer Genterra, Russia) was non-optimized. We have made our own buffer (the composition of buffer is given in Materials and methods) and both the reproducibility of Cts and reaction rate of LAMP have improved significantly. So we have repeated the experiments and achieved a better result for the fLAMP sensitivity experiment.*

1- 5. There are multiple issues with the diagnostic sensitivity/specificity analyses. In order to assess the diagnostic sensitivity/specificity of LAMP assays, you are comparing to the results from RT-PCR assay as the 'gold-standard' (p. 8). It is important to indicate what the cutoffs are for positive/negative specimens on the RT-PCR assay to inform how Ct values in the RT-PCR assay reflect viral genome quantity. Moreover, Table 3 and 4 are confusing / not informative. I would recommend making tables that demonstrate positive and negative results across the gold-standard versus novel method of comparison (e.g., cLAMP or fLAMP). Furthermore, agreement analyses are worthwhile (e.g., Cohen's kappa) to determine the utility of the LAMP assays (in addition to specificity and sensitivity statistical analyses. If still of interest, linear regressions of RT-PCR versus c/fLAMP Ct values can be performed to compare.

Author's answer *We changed table 3 that demonstrate positive and negative results across the gold-standard versus cLAMP and fLAMP Table 4 has changed too as the degree of agreement (Cohen's kappa) between PCR and LAMP assays was calculated.*

1-6. The comparison of automated versus manual DNA extraction methods for the gold-standard RT-PCR method (p. 10) is irrelevant for a study that compares novel LAMP diagnostic methods. If anything, extraction methods should be compared among each of the cLAMP and fLAMP methods.

Author's answer *We compared the nucleic acid extraction methods before LAMP assays developing. We want to show that automatic extraction can remain the quality of NA extraction but reduce the time of testing.*

If you consider it necessary to conduct the same research for LAMP assays, we are ready, as the material of fecal extracts is stored in our collection. But this will require additional time.

Minor 1-1 For those unfamiliar with the RT-PCR assay utilized as the 'gold-standard', please indicate whether the assay can detect and differentiate between other adenovirus subtypes. Moreover, please indicate what cutoffs are used by the clinical laboratory (or in accordance with manufacturer's recommendations) to define positive or negative specimens as well as whether specimens are run as singletons or in replicates.

Author's answer *We take into account your comments in "Materials and methods" "AmpliSens® OKI-screen-FL kit specifically detects adenovirus F and does not detect other types of adenoviruses. Declared analytical sensitivity for adenovirus F DNA isolated from fecal extract– up to 10⁴ copies/ml."*

Minor 1-2. On p. 4-5, please indicate how the cLAMP assay is read out. Is this qualitatively read by eye or by a colorimeter? If by the former, it would be helpful to discuss the reliability of read-out by eye particularly by various users.

Author's answer *We take into account your comments in "Materials and methods" The results were independently visually evaluated by two persons and were documented using photo.*

Minor 1-3 On p. 8, please elaborate what the Ct values were of the mastadenoviruses and bacteria tested in the specificity assay. This will depend on what your cutoff is for your defining positive/negative results.

Author's answer *We haven't detect positive signal (color change or Ct values) for other mastadenoviruses and bacteria. We add these data to manuscript (fig 3, 4).*

Minor 1-4

- **Italicize genus and species names of pathogens tested for specificity (e.g., p. 6).**

OK

- **"Primer sensitivity" is misleading as BLAST-ing sequences does not inform sensitivity (e.g., lowest concentration of detection) of primers/target/assay (p. 7).**

Author's answer *we changed the phrase to "The maximum number of database sequences of the human adenovirus F hAdV40/41 groups was BLAST screened to make sure that sequence of primers have high identity to target region of hAdV40/41 genomes."*

- Please go through the manuscript to ensure English grammar is intact/appropriate.

Author's answer I did it as far as my knowledge of the language allowed me. Thank you for the detailed examine of the manuscript and valuable remarks

Reviewer #2 (Comments for the Author):

The manuscript by Malova describes a LAMP assay for enteric adenoviruses in feces of children. The manuscript is lacking in details in several aspects detailed below:

2-1 Abstract should reflect the targets in this assay (hexon gene of adenovirus).

Author's answer *We have changed the Abstract : "...We designed colorimetric LAMP (cLAMP) and real-time LAMP (fLAMP) with fluorescent probes to detect the DNA of adenoviruses F hAdV40/41 hexon gene..."*

2-2 Provide clarity regarding primer design. For example in Table 1. it would be useful to indicate which ones are probes. Also indicate which target is being detected.

Author's answer *We have changed the Table 1 to include new column "oligonucleotide" to specify primers - forward outer primer, forward inner primer, reverse outer primer, reverse inner primer, loop primer, loop primer, forward inner primer, reverse inner primer, reverse inner primer, probe / loop primer, quencher correspondently.*

2-3 How was specificity of primers and probes tested?

Author's answer *LAMP target-specific primers specific for hAdV40/41 were designed on unique template regions of hAdV40/41 hexone genes that are not similar to hexone genes of Human mastadenovirus A, B, C, D, G, Bat mastadenovirus, Canine mastadenovirus, Deer mastadenovirus, Dolphin mastadenovirus, Murine mastadenovirus, Ovine mastadenovirus, Porcine mastadenovirus, Simian mastadenovirus and other taxons. Representative sequences of these taxons were aligned in Mega X. Also we used the search tool NCBI BLAST it was shown the primers sequences have not high identity to other genomes from GeneBank. The specificity of the LAMP assays (no cross-reactivity with other pathogens) was evaluated in the experiments using strains of human mastadenovirus, enterobacteria and Adenovirus F PCR negative fecal DNA extract.*

2- 4. Table 5. How was discrepancy resolution performed for the 2 samples that were positive after automated extraction?

Author's answer *The discordant results were confirmed by repeated tests (PCR and LAMPs). Unfortunately, Sanger sequencing of hexone gene for 2 samples that were positive after automated extraction was impossible as the virus load was very low.*

2-5. Both results and discussion section can be clarified by including definite details on what the targets were for each assay and what the comparator assay was. As currently written, it is unclear whether assay for all targets including rotavirus and norovirus were in this study or whether this was a commercial assay that the LAMP assay was being compared to?

Author's answer *The samples of the clinical panel (107 fecal extracts) were PCR tested for Adenovirus F presence. Real-time PCR was performed using an AmpliSens® OKI-screen-FL kit (Central Research Institute of Epidemiology, Russia) in CFX96 RealTime PCR machines (Bio-Rad, USA). Adenovirus F-positive samples were selected and tested by AmpliSens® OKI-screen-FL kit for the presence of other enteric viral and bacterial pathogens (rotavirus A, norovirus 2, astrovirus Shigella spp., E. coli (EIEC), Salmonella spp., and Campylobacter spp.), included in OKI-screen-FL test. These experiments were carried out for estimation amount of incidences of mixed infections among children with acute adenovirus gastroenteritis.*

2-6. A brief explanation on how this assay can be utilized clinically will be valuable. It is clear that specimens with low viral load may not be detected by this assay. Is there a way that the rapidity of the test can be utilized in clinical decision making that would still make the test useful despite its insensitivity? Especially since it is low cost?

Author's answer To clarify the significance of Colorimetric LAMP we added a sentence In "Discussion" - Colorimetric LAMP allows visual detection in a short time, so this method is suitable even for budget-restricted laboratories in the absence of expensive PCR machines.

Thank you for the detailed examine of the manuscript and valuable remarks

Reviewer #3 (Comments for the Author):

Shuryaeva et al. report on the development of a LAMP assay for detecting adenovirus F DNA from patient clinical samples. The authors report high sensitivity and specificity. The authors demonstrate high performance on a panel of fecal samples collecting from children experiencing acute viral gastroenteritis. In several locations in the manuscript, the validation could use more description or development.

Major Comments:

3-1 There is no mention of ethical approval for the study. Was patient consent received for using clinical samples for research purposes? Patient clinical features were also mentioned ("Coinfected patients had lower base excess").

Ethical approval for using clinical samples for research purposes was obtained from the local ethics committee at Children's City Clinical Hospital No. 9, Moscow, Russia (protocol № 44, 19.04.22).

3-2 Figure 1, panel 10³ copies/mL appears to be manipulated and represents two separate experiments. Can the authors please clarify? If the samples with equal concentration were run on different strips that is fine but the images shouldn't be edited to look as if they are one continuous image (and I would appreciate seeing the raw, unedited images). If the authors have more than three replicates for each concentration, can that additional data be shown? Six versus three replicates would greatly improve the interpretability of the sensitivity experiments as only three samples is insufficient. In general, I strongly recommend the authors increase the numbers of replicates at concentrations close to the cut-off value of LAMP performance and perform probit regression to obtain 95% and 50% limit of detection estimates.

We repeated the experiment with six replicates at concentrations close to the cut-off value of colorimetric c-LAMP performance and submit the raw image as one continuous image. We increased the numbers of replicates at concentrations close to the cut-off value of LAMP performance to six for f-LAMP with assimilating FRET primer probe too.

3-3 The claim that cLAMP and fLAMP have comparative sensitivity with PCR, the authors should demonstrate the sensitivity of PCR targeting the same region. This is also hard to interpret along the section "Among the 51 PCR-positive adenovirus F samples, 31 samples had a sufficient viral load with Ct PCR threshold cycles of less than 36, which corresponds to a virus content of 10³ copies/ml or more. The 20 PCR-positive adenovirus F samples had threshold cycles of more than 36, which corresponds to a very low virus content in these samples". If the PCR test was producing positive results for a higher proportion of samples with a "virus content" less than 10³ copies/ml than cLAMP or fLAMP surely that means PCR is more sensitive?

When we claim that cLAMP and fLAMP have sensitivity comparative to PCR, we mean that 10³ copies/ml (10 copies per reaction tube) are potentially good value for PCR assays. The aim of the work is the development of LAMP assay. We use AmpliSensOKI-screen kit for in vitro diagnostics to compare diagnostic sensitivity LAMP and PCR (not to evaluate the analytical sensitivity). Thus, we found that PCR is more sensitive although the detection limit of LAMP assays measured on plasmid DNA was good (10³ copies/ml). Declared detection limit of PCR AmpliSensOKI-screen assay is 10⁴ copies/ml.

May you permit me to stay without changes phrases regarding the diagnostic sensitivity of assays in the manuscript?

3-4 Please provide the quantification of other targets used when testing the assay specificity. Were these tested on both cLAMP and fLAMP?

Yes, the specificity of LAMP for hAdV40/41 adenoviruses was estimated for both cLAMP and fLAMP assays (fig. 3,4). To determine the specificity of colorimetric c-LAMP and f-LAMP the reaction time was increased to 60 minutes for both. We pointed in the manuscript the titers of the virus strains (3.0 log TCID 50/ml - 5.0 log TCID 50/ml) in Table 2 and added information about bacterial strains concentrations (1×10^6 CFUs/ml) also.

3-5 The phrase "Additionally, the f-LAMP assay was able to detect several pathogens in one multiplexed reaction." is unclear. Did the authors develop a multiplexed LAMP reaction? That would be a strong development and would increase the utility of the technical advancements presented in this article.

No, at the moment we have not developed a multiplexed LAMP assay to detect several pathogens. Therefore, we only point out the possibility of such a technique.

Minor

3- 6 Please define the acronyms LAMP, PCR, and Ct in the main text of the article.

ok

3-7 In the methods, the authors state the samples were tested on serial dilutions from 5×10^2 copies/ml to 10^6 copies/ml. The results section states that cLAMP was tested from 10^2 copies/mL to 10^4 copies/mL and the fLAMP studies only show to 10^4 copies/mL. Please clarify.

It was a mistake and it has been fixed in the text of the manuscript.

3-8 The inclusion of "Ct" values for fLAMP is confusing as the PCR machine isn't really "cycling". It may result in improper comparison with qPCR methods. I would remove the Ct column from column 2 and only leave the time column.

I deleted the "Ct" column" in the tables of the manuscript and named the remain column as " Ct/2,min"

Thank you for the detailed examine of the manuscript and valuable remarks

May 20, 2022

Dr. Ekaterina Evgenevna Davydova
Federal State Budgetary Institution Centre for Strategic Planning and Management of Biomedical Health Risks of the Federal Medical Biological Agency
Pogodinskaya Str, 10
Moscow 119121
Russia

Re: Spectrum00516-22R1 (Development and application of LAMP assays for the detection of enteric adenoviruses in feces.)

Dear Dr. Ekaterina Evgenevna Davydova:

Your revised manuscript was re-reviewed by two of the original reviewers. Both reviewers still require modifications before publication and both reviewers are in agreement that the information regarding the specificity analyses is inadequate. Generally, it is the policy of Microbiology Spectrum not to invite a second revision. However, given that this is a topic of interest to readership in the setting of global adenovirus 41 hepatitis in children, I will make an exception. Please address the reviewers specific questions clearly and in detail in the Methods. The manuscript cannot move forward for publication without attention to those requests.

Link Not Available

Sincerely,

Karen Carroll

Journals Department
Reviewer comments:

Reviewer #2 (Comments for the Author):

Revisions have addressed some of the comments. A few issues:

1. Analytical sensitivity was measured using a plasmid which does not completely address the performance on clinical specimens, were adenovirus control material from commercial sources tried for the complete extraction and amplification

performance evaluation?

2. Was specificity of primers checked using other commonly encountered viruses such as rotavirus and norovirus? Lines 197-201 indicate checking cross reactivity to Salmonella, shigella and other human mastadenovirus.

3. Manuscript could benefit from a spelling and grammar check

Reviewer #3 (Comments for the Author):

I thank the authors very much for revising the manuscript to address the points raised by myself and the other reviewers and I do believe the manuscript is stronger now. I appreciate that the demand for rapid tests for adenoviral infection has risen given the fulminant hepatitis cases globally. For this study, I have some remaining questions.

Major comments

- The ethical approval obtained is still not clear. By the methods, it appears as if the application for ethical approval was granted (19.4.2022) after the study was completed (first manuscript was received February 2022). Can the authors please clarify? The document providing the ethical approval would be helpful as retrospective ethical approval is generally inadvisable. ASM does not appear to have clear guidelines on retrospective ethical approval and COPE guidelines are also equivocal.

- The section on analytical specificity is still not clear. Providing the viral and bacterial titers of the stocks does not specify the quantity used as input into the LAMP assay. Please specify the starting volume extracted, elution volume, and volume spiked in to the assay. Were the mastadenovirus samples spiked into fecal extracts before extraction or extracted from stock cultures Was the eluate diluted in water/PBS and then spiked into the LAMP assay? These details would allow a researcher to roughly estimate the genome copies/ml.

Minor comments

- Page 2, line 63: replace "thermostat" with "thermocycler".

- Page 7, lines 205-209: This is a repeat of page 3, lines 81-85.

- Page 12, lines 283-285: This content is present in the methods.

- Page 14, lines 329-330: The authors have not removed the phrase "Additionally, the f-LAMP assay was able to detect several pathogens in one multiplexed reaction." The "response to reviewer comments" document clarified that they have not demonstrated this capability. The sentence needs to be modified or removed.

Staff Comments:

Preparing Revision Guidelines

Please return the manuscript within 60 days; if you cannot complete the modification within this time period, please contact me. If you do not wish to modify the manuscript and prefer to submit it to another journal, please notify me of your decision immediately so that the manuscript may be formally withdrawn from consideration by Microbiology Spectrum.

Dear Reviewers,

Thank you very much for repeated corrections of our manuscript. We have answered to your comments in a point-by-point manner below.

Changes to the manuscript (Marked-Up_Manuscript_2022-06-06_Spectrum00516-22R2.docx) are shown in red or yellow. Also the text of the manuscript was changed in accordance with journal requirements so the section "Materials and methods" was moved down.

Sincerely yours,

Ekaterina Davydova

Reviewer #2 (Comments for the Author):

Major comments

#2-1 Analytical sensitivity was measured using a plasmid which does not completely address the performance on clinical specimens, were adenovirus control material from commercial sources tried for the complete extraction and amplification performance evaluation?

We have strains of Human mastadenovirus B/C/D heterologous groups but unfortunately, have no strains of Human mastadenovirus F. Adenovirus F DNA positive samples were obtained from fecal extract which confirmed by AmpliSens® OKI-screen-FL in vitro diagnostics kit . Some of the PCR positive samples were confirmed by Sanger sequencing of the hexon gene fragment too. This was the reason for using a pure plasmid DNA with a certain concentration. This approach for measuring sensitivity of PCR is widely used.

The information about the efficiency of nucleic acid extraction of in vitro diagnostics kit AmpliTest RiboPrep is given in a manual of the kit (the information is posted on our website www.amplitest.ru). So the efficiency of nucleic acid extraction using AmpliTest RiboPrep kit is at least 50%. It was estimated using model clinical samples (sputum, smears and fecal extract) contains SARS CoV-2 with known concentration.

PrintScreens of Amplifest Ribo Prep kit manual, www.amplitest.ru

производства ФГБУ «ЦСП» ФМБА России (Р.У. № РЗН 2020/9765).

Эффективность экстракции

Таблица 2

Наименование показателя	Норма
Соотношение концентраций РНК SARS CoV-2 в модельном образце биоматериала прошедшем экстракцию и контрольном образце без экстракции.	Эффективность экстракции не менее 50 %

Примечание. При анализе с помощью системы QX100 droplet digital PCR для проведения капельной цифровой ПЦР (Р.У. № ФСЗ 2012/13278)

Г.А. Шапугин
2020 г.

ИНСТРУКЦИЯ ПО ПРИМЕНЕНИЮ
комплекта реагентов для экстракции РНК/ДНК
из биологического материала
«АмплиТест РИБО-преп»

<https://amplitest.ru/%d0%b8%d0%bd%d1%81%d1%82%d1%80%d1%83%d0%ba%d1%86%d0%b8%d1%8f-%d0%b0%d0%bc%d0%bf%d0%bb%d0%b8%d1%82%d0%b5%d1%81%d1%82-%d1%80%d0%b8%d0%b1%d0%be-%d0%bf%d1%80%d0%b5%d0%bf-19-08-20/>

#2-2 Was specificity of primers checked using other commonly encountered viruses such as rotavirus and norovirus? Lines 197-201 indicate checking cross reactivity to Salmonella, shigella and other human mastadenovirus.

We use panel of DNA pathogens causing intestinal infections and human mastadenoviruses related to adenovirus F for specificity testing. We took into account that rotavirus and norovirus are RNA viruses so their genomes are not amplified in LAMP without Reverse Transcriptase Activity.

It is the reason why we do not check the specificity of primers to rotavirus and norovirus RNA. Also non-specific LAMP amplification was not observed for PCR rotavirus and norovirus positive clinical samples which were PCR-negative for adenovirus.

#2-3. Manuscript could benefit from a spelling and grammar check

Reviewer #3

Major comments

#3-1 The ethical approval obtained is still not clear. By the methods, it appears as if the application for ethical approval was granted (19.4.2022) after the study was completed (first manuscript was received February 2022). Can the authors please clarify? The document providing the ethical approval would be helpful as retrospective ethical approval is generally inadvisable. ASM does not appear to have clear guidelines on retrospective ethical approval and COPE guidelines are also equivocal.

The protocol No. 44 of the Ethics Committee at the G.N. Speranskiy Children Hospital No 9 dated April 19, 2022 is attached to manuscript account. The Ethics Committee have made amendments to the protocol №39 dated 26.10.2021z which concern the research project "Comparative laboratory study on the etiological agents identification of acute respiratory diseases/acute gastroenteritis from smears / feces samples». In accordance with protocol №39 dated 26.10.2021z the laboratory of the G.N. Speranskiy Children Hospital was permitted to use the clinical material from children for scientific research (page 4 of protocol No. 44).

After your request we asked The Ethics Committee at the G.N. Speranskiy Children Hospital No 9 made amendments to the Decision (protocol №39 dated 26.10.2021z) to permit using of the depersonalized samples for scientific research the laboratory of the G.N. Speranskiy Children Hospital in collaboration with Federal State Budgetary Institution Centre for Strategic Planning and Management of Biomedical Health Risks.

#3-2 - The section on analytical specificity is still not clear. Providing the viral and bacterial titers of the stocks does not specify the quantity used as input into the LAMP assay. Please specify the starting volume extracted, elution volume, and volume spiked in to the assay. Were the mastadenovirus samples spiked into fecal extracts before extraction or extracted from stock cultures Was the eluate diluted in water/PBS and then spiked into the LAMP assay? These details would allow a researcher to roughly estimate the genome copies/ml.

Corrections were made to the text of the manuscript at

Page 10 lines 252-253; Page 11 lines 259-260; Page 14 lines 361-367

Minor comments

#3-3 Page 2, line 63: replace "thermostat" with "thermocycler".

I removed the word "thermostat" (Page 3, line 71) . The word is not essential in the text. But I mean, a thermostat is a device that can heat up to a setpoint temperature and cannot thermocycle.

#3-4 Page 7, lines 205-209: This is a repeat of page 3, lines 81-85.

The repeated phrase was removed

#3-5 Page 12, lines 283-285: This content is present in the methods.

The repeated phrase was removed

#3-6 Page 14, lines 329-330: The authors have not removed the phrase "Additionally, the f-LAMP assay was able to detect several pathogens in one multiplexed reaction." The "response to reviewer comments" document clarified that they have not demonstrated this capability. The sentence needs to be modified or removed.

The phrase was removed

МИНИСТЕРСТВО ЗДРАВООХРАНЕНИЯ РОССИЙСКОЙ ФЕДЕРАЦИИ

Департамент здравоохранения города Москвы

Этический комитет при ГБУЗ «ДГКБ №9 им. Г.Н. Сперанского ДЗМ»

123317, г. Москва, Шмитовский проезд, 29, Тел. (499) 256-21-62

Выписка из протокола заседания №44

Этического комитета при ГБУЗ "ДГКБ №9 им. Г.Н. Сперанского ДЗМ"

от "19" апреля 2022 года

Полное наименование и место нахождения: Этический комитет при ГБУЗ "ДГКБ №9 им Г.Н. Сперанского ДЗМ" г. Москва, Шмитовский проезд д. 29.Тел. (499)256-21-62

Форма проведения: очередное заседание членов Этического комитета (совместное присутствие членов для обсуждения вопросов повестки дня и принятия решений по вопросам, поставленным на голосование)

Дата проведения: от 19.04.2022 года

Место проведения: г. Москва, Шмитовский проезд, 29, Корпус 1, лекционный (красный) зал

Время проведения: 12.00

Председатель заседания: Гусева Н.Б.

Секретарь заседания: Аминова А.И.

На заседании присутствовали Члены Комитета:

№	ФИО	Должность	Право голоса	Присутствие
1.	Гусева Наталья Борисовна Председатель ЛЭК	Д.м.н., профессор, руководитель МГЦ детской урологии, андрологии и патологии тазовых органов «ДГКБ №9 им. Г.Н. Сперанского», главный научный сотрудник отдела хирургии детского возраста ФГБОУ ВО РНИМУ им. Н.И. Пирогова, главный научный сотрудник НПЦ детской психоневрологии ДЗМ	есть	Лично
2.	Крапивкин Алексей Игоревич	Д.м.н., заместитель главного врача по медицинской части	есть	Лично

	заместитель председателя ЛЭК			
3.	Аминова Альфия Иршадовна Ответственный Секретарь ЛЭК	Д.м.н., профессор кафедры пропедевтики детских болезней Первого МГМУ им. И.М. Сеченова, врач-гастроэнтеролог	есть	Лично
4.	Левина Дарья Михайловна Технический Секретарь ЛЭК	Аспирант, старший лаборант кафедры педиатрии и детских инфекционных болезней Первого МГМУ им И.М.Сеченова, врач-педиатр в отделении аллергологии-иммунологии №1	есть	Лично
5.	Будкевич Людмила Иасоновна	Д.м.н., профессор, зав. ожоговым Центром, заведующая 2 ожоговым отделением, врач-детский хирург	есть	Лично
6.	Батаев Саидхасан Магомедович	Д.м.н. детский хирург, главный научный сотрудник НИИ хирургии детского возраста ГБОУ ВПО РНИМУ им. Н.И. Пирогова	Право голоса	Лично
7.	Горшкова Лариса Юрьевна	Заместитель Директора благотворительного фонда «Детская больница»	есть	Лично
8.	Долгушина Ольга Анатольевна	Заведующая приемным отделением (педиатрическое, инфекционное)	есть	Лично
9.	Елков Андрей Юрьевич	К.м.н., заведующий отделением ультразвуковой диагностики, врач ультразвуковой диагностики	есть	Лично
10.	Рыжов Евгений Александрович	К.м.н., заместитель главного врача по хирургии	есть	Лично
11.	Морено Илья Геннадьевич	К.м.н., заведующий кардиологическим отделением, главный внештатный специалист детский кардиолог ЦАО ДЗМ, врач детский кардиолог	есть	Лично
12.	Кондратенко Наталья Владимировна	Врач клинический фармаколог	есть	Лично
13.	Коновалов Александр Карпович	Д.м.н., заведующий отделением гнойной хирургии (1 х о), врач-детский хирург	есть	Лично

14.	Корсунский Илья Анатольевич	Д.м.н., заведующий КДЦ детской иммунологии и аллергологии, врач аллерголог-иммунолог, доцент кафедры педиатрии и детских инфекционных болезней Клинического института детского здоровья им. Н.Ф. Филатова ФГАОУ ВО 1 МГМУ им. И.М. Сеченова	есть	Лично
15.	Лазарев Владимир Валентинович	Д.м.н., профессор кафедры педиатрии и детских инфекционных болезней Клинического института детского здоровья им. Н.Ф. Филатова ФГАОУ ВО 1 МГМУ им. И.М. Сеченова МЗ России. Главный специалист больницы по инфекционным заболеваниям, врач-инфекционист, врач-педиатр. Руководитель Университетской клиники педиатрии и детских инфекционных болезней 1 МГМУ им. И.М. Сеченова	есть	Лично
16.	Ивойлов Алексей Юрьевич	Д.м.н., заведующий оториноларингологическим отделением, главный детский оториноларинголог г.Москвы	Право голоса	Лично
17.	Галеева Елена Валентиновна	Заведующая лабораторией, врач клинической лабораторной диагностики	Право голоса	Лично
18.	Мунблит Даниил Борисович	К.м.н., профессор кафедры педиатрии и инфекционных болезней Первого МГМУ им. И.М.Сеченова	Право голоса	Лично
19.	Продеус Андрей Петрович	Д.м.н., профессор, врач аллерголог-иммунолог, главный педиатр ДГКБ №9	Право голоса	Отсутствовал

Число голосов, принадлежащих членам Комитета, принявшим участие в заседании по вопросам повестки дня - "18".

Кворум для проведения заседания Комитета имеется. Заседание Комитета признано правомочным принимать решения по всем вопросам повестки дня.

Этический комитет на своем заседании в соответствии с правилами GCP рассмотрел возможность внесения поправок в протокол научно-исследовательской работы «Сравнительное лабораторное исследование по Регистрационному удостоверению на медицинское изделие № ФСР 2011\11258 и № ФСР 2008/02265 по этиологической расшифровке ОРВИ и ОКИ соответственно в мазках из респираторного тракта и в кале с помощью ПЦР в режиме реального времени с дальнейшей передачей оставшегося после исследования биоматериала в «ЦСП Минздрава» с целью разработки новой диагностической панели

Название: Сравнительное лабораторное исследование по Регистрационному удостоверению на медицинское изделие № ФСР 2011\11258 и № ФСР 2008/02265 по этиологической расшифровке ОРВИ и ОКИ соответственно в мазках из респираторного тракта и в кале с помощью ПЦР в режиме реального времени с дальнейшей передачей оставшегося после исследования биоматериала в «ЦСП Минздрава» с

целью разработки новой диагностической панели.

Исследование одобрено на заседании ЛЭК от 26.10.2021г. Документы предоставлены.

Главный исследователь: Гордукова Мария Александровна, клинико-диагностическая лаборатория, группа ПЦР, ГБУЗ «Детская городская больница №9 им. Г.Н. Сперанского ДЗМ», биолог, 8-926-291-38-65, ma.gordukova@dgkb-9.ru

Документы предоставлены.

Вопросы к главному исследователю не было.

Вопрос, поставленный на голосование: возможность одобрения внесения поправок в протокол научно-исследовательской работы

Итоги голосования:

«За»		«Против»		«Воздержался»
Количество голосов	%от общего число голосов	Количество голосов	%от общего число голосов	Количество голосов
18	100	0	0	0

Принятое решение по рассматриваемому вопросу: Одобрить внесение поправок в протокол научно-исследовательской работы Сравнительное лабораторное исследование по Регистрационному удостоверению на медицинское изделие № ФСР 2011\11258 и № ФСР 2008/02265 по этиологической расшифровке ОРВИ и ОКИ соответственно в мазках из респираторного тракта и в кале с помощью ПЦР в режиме реального времени с дальнейшей передачей оставшегося после исследования биоматериала в «ЦСП Минздрава» с целью разработки новой диагностической панели.

Председатель

 /Гусева Н.Б./

Секретарь

 /Аминова А.И./

Тех.секретарь

 /Левина Д.М./

19.04.2022г.

June 7, 2022

Dr. Ekaterina Evgenevna Davydova
Federal State Budgetary Institution Centre for Strategic Planning and Management of Biomedical Health Risks of the Federal Medical Biological Agency
Pogodinskaya Str, 10
Moscow 119121
Russia

Re: Spectrum00516-22R2 (Development and application of LAMP assays for the detection of enteric adenoviruses in feces.)

Dear Dr. Ekaterina Evgenevna Davydova:

Your manuscript has been accepted, and I am forwarding it to the ASM Journals Department for publication. You will be notified when your proofs are ready to be viewed.

Sincerely,

Karen Carroll
Editor, Microbiology Spectrum
